Surface display for metabolic engineering of industrially important acetic acid bacteria

Blank Marshal 1
Schweiger Paul 2 pschweiger@uwlax.edu
1 Biology Department, Missouri State University , Springfield, MO , USA
2 Department of Microbiology, University of Wisconsin-La Crosse , La Crosse, WI , USA
Winkler Robert
Electronic publication date: 2018 Apr 6
Publication date: 2018
Volume: 6
Electronic Location ID: e4626
Received 2018 Feb 21; Accepted 2018 Mar 26
Copyright: © 2018 Blank and Schweiger
Copyright year: 2018
Copyright holder: Blank and Schweiger
License: This is an open access article distributed under the terms of the Creative Commons Attribution License, which permits unrestricted use, distribution, reproduction and adaptation in any medium and for any purpose provided that it is properly attributed. For attribution, the original author(s), title, publication source (PeerJ) and either DOI or URL of the article must be cited.
License URL: https://creativecommons.org/licenses/by/4.0/

Keywords: Surface display, Fusion linkers, Biocatalysis, Outer membrane proteins, Gluconobacter oxydans

Funding: Missouri State University Biology Department and the Graduate College This work was supported by funds from the Missouri State University Biology Department and the Graduate College. The funders had no role in study design, data collection and analysis, decision to publish, or preparation of the manuscript.

==============================
Acetic acid bacteria have unique metabolic characteristics that suit them for a variety of biotechnological applications. They possess an arsenal of membrane-bound dehydrogenases in the periplasmic space that are capable of regiospecific and enantioselective partial oxidations of sugars, alcohols, and polyols. The resulting products are deposited directly into the medium where they are easily recovered for use as pharmaceutical precursors, industrial chemicals, food additives, and consumer products. Expression of extracytoplasmic enzymes to augment the oxidative capabilities of acetic acid bacteria is desired but is challenging due to the already crowded inner membrane. To this end, an original surface display system was developed to express recombinant enzymes at the outer membrane of the model acetic acid bacterium Gluconobacter oxydans. Outer membrane porin F (OprF) was used to deliver alkaline phosphatase (PhoA) to the cell surface. Constitutive high-strength p264 and moderate-strength p452 promoters were used to direct expression of the surface display system. This system was demonstrated for biocatalysis in whole-cell assays with the p264 promoter having a twofold increase in PhoA activity compared to the p452 promoter. Proteolytic cleavage of PhoA from the cell surface confirmed proper delivery to the outer membrane. Furthermore, a linker library was constructed to optimize surface display. A rigid (EAAAK)1 linker led to the greatest improvement, increasing PhoA activity by 69%. This surface display system could be used both to extend the capabilities of acetic acid bacteria in current biotechnological processes, and to broaden the potential of these microbes in the production of value-added products.

Introduction

Gluconobacter oxydans is an industrially important microbe belonging to the family Acetobacteriaceae, commonly referred to as the acetic acid bacteria. G. oxydans is an incomplete oxidation specialist known for its ability to partially oxidize alcohols, polyols, and monosaccharides to produce a diverse array of aldehydes, ketones, and organic acids. These reactions occur in the periplasmic space, as G. oxydans possesses an arsenal of dehydrogenases bound to the inner membrane. Therefore, industrially valuable compounds are excreted directly into the growth medium, where they are easily obtained for use as food additives, pharmaceutical precursors, industrial chemicals, and consumer products (Deppenmeier & Ehrenreich, 2009; Deppenmeier, Hoffmeister & Prust, 2002; Prust et al., 2005). What is more, the membrane-bound dehydrogenases are both regiospecific and enantioselective, allowing the microorganism to produce chiral compounds from precursors containing multiple identical functional groups, such as sugars and polyols. Additionally, the membrane-bound dehydrogenases contain prosthetic groups that channel electrons to ubiquinone in the respiratory chain, which facilitates rapid oxidation of substrates (Deppenmeier & Ehrenreich, 2009; Prust et al., 2005). G. oxydans is also osmotolerant and acidophilic, which are desirable industrial characteristics (Olijve & Kok, 1979).

Acetic acid bacteria serve many roles in biotechnology, but G. oxydans is particularly important (Deppenmeier, Hoffmeister & Prust, 2002; Raspor & Goranovic, 2008). Currently, G. oxydans is used to produce l-sorbose, the precursor to vitamin C (Pappenberger & Hohmann, 2014; Yang & Xu, 2016) and 6-amino-6-desoxy-l-sorbose, the precursor to the antidiabetic drug miglitol (Schedel, 2000). Additionally, this bacterium is used to produce, dihydroxyacetone and erythrulose, which are primarily used as tanning agents in cosmetics (De Muynck et al., 2007; Voss, Ehrenreich & Liebl, 2010), as well as gluconate and gluconate derivatives, which serve as sequestering agents and drug precursors (De Muynck et al., 2007; Deppenmeier, Hoffmeister & Prust, 2002). Despite their widespread use in industry, progress toward metabolic engineering of acetic acid bacteria has been partially limited by the lack of molecular tools available for genetic manipulation of this group of microbes (Kallnik et al., 2010). Currently, constitutive expression vectors are available for gene expression (Kallnik et al., 2010; Shi et al., 2014; Zhang et al., 2010), and two markerless deletion systems have been developed (Kostner et al., 2013; Peters et al., 2013). The ability to produce recombinant enzymes for the modification of extracellular substrates has the potential to expand and compliment the natural incomplete-oxidative metabolism of acetic acid bacteria. Signal peptides that allow for periplasmic export are known (Kosciow et al., 2014), but the ability to produce additional enzymes bound to the cytoplasmic membrane is limited because this space is already crowded by the large number of native dehydrogenases (Guigas & Weiss, 2016). Production of extracellular enzymes is also challenging because G. oxydans lacks the machinery to secrete proteins across the outer membrane (Prust et al., 2005).

To overcome these limitations, a surface display system for expression of recombinant enzymes at the cell surface of G. oxydans and other acetic acid bacteria was designed. This molecular tool enables production of active enzymes with access to the extracellular space, bypassing the crowded inner membrane. Surface display involves translational fusion of a passenger protein with an anchor that innately localizes to the cell surface. Surface display offers several advantages for biocatalysis. Most importantly, substrates do not have to cross membrane barriers to interact with recombinant enzymes, and resulting products are deposited directly into medium, permitting simplified extraction without the need for cell lysis. Additionally, cells can be used for multiple rounds of biocatalysis, as they can be removed from a spent reaction by centrifugation and then resuspended in a solution containing new substrate. Lastly, surface display eliminates the need to purify enzymes (Schüürmann et al., 2014). To this end, a truncated version of outer membrane porin F (OprF) from Pseudomonas aeruginosa was translationally fused to alkaline phosphatase (PhoA), from Escherichia coli, and activity of the reporter enzyme was quantified in a whole-cell assay. PhoA was proteolytically cleaved from the cell, demonstrating that it was properly delivered to the outer leaflet of the outer membrane. Finally, the surface display system was optimized by testing the effects of various linkers on biocatalysis.

Methods

Strains and media

Pseudomonas aeruginosa PAO1 (DSMZ 22644) was grown in tryptic soy broth (Becton Dickinson, Franklin Lakes, NJ, USA). E. coli 10β (New England Biolabs, Ipswich, MA, USA), herein E. coli, was grown in lysogeny broth (1% tryptone, 0.5% yeast extract, 1% NaCl) with 100 μg/mL streptomycin added for strain maintenance. G. oxydans 621H, hereafter G. oxydans, was grown in yeast mannitol (YM) broth composed of 2% mannitol and 0.6% yeast extract, with 50 μg/mL cefoxitin added for strain maintenance. Plasmids were maintained by addition of 50 μg/mL kanamycin. Electrocompetent G. oxydans was prepared by growth in electroporation medium (Kallnik et al., 2010). After electroporation, G. oxydans cells were plated on yeast glucose calcium carbonate (YGC) agar (2% glucose, 1.5% agar, 0.7% CaCO3, and 0.6% yeast extract) containing cefoxitin and kanamycin.

Materials and molecular techniques

Standard molecular techniques were done according to manufacturer’s protocols. Plasmid DNA was extracted using a GeneJet Plasmid Miniprep kit (ThermoFisher Scientific, Waltham, MA, USA) and genomic DNA was extracted using a GenElute Bacterial Genomic DNA kit (Millipore-Sigma, St. Louis, MO, USA). Phusion DNA polymerase and DreamTaq polymerase, FastDigest restriction enzymes, and T4 ligase were purchased from ThermoFisher Scientific (Waltham, MA, USA). Factor Xa protease was purchased from New England Biolabs (Ipswich, MA, USA). Primers were purchased from either Eurofins Genomics (Louisville, KY, USA) or Integrated DNA Technologies (Coralville, IA, USA) (Table 1). DNA sequencing was done by Eurofins Genomics (Louisville, KY, USA).

Table 1 Plasmids and primers used for molecular cloning.

Plasmid or primer	Description or sequencea	Source or restriction site	
Plasmids	
pBBR1p264-ST	pBBR1p264 derivative containing a Strep-tag (ST) sequence	Zeiser et al. (2014)	
pBBR1p452-ST	pBBR1p452 derivative containing a ST sequence	Kallnik et al. (2010)	
pBBR1p264-oprF-ST	pBBR1p264-ST derivative expressing oprF188 from P. aeruginosa	This study	
pBBR1p452-oprF-ST	pBBR1p452-ST derivative expressing oprF188 from P. aeruginosa	This study	
pBBR1p264-oprF-phoA	pBBR1p264-oprF-ST derivative with ST removed, expressing phoA from E. coli	This study	
pBBR1p452-oprF-phoA	pBBR1p452-oprF-ST derivative with ST removed, expressing phoA from E. coli	This study	
pBBR1p452-oprF-CL-phoA	pBBR1p452-oprF-phoA derivative with phoA removed, replaced by CL-phoA encoding a Factor Xa cleavable linker	This study	
pBBR1p264-oprF-FL1-phoA	pBBR1p264-oprF-ST derivative with oprF-ST removed, replaced by oprF-FL1 and FL1-phoA encoding a (GGGGS)1 flexible linker	This study	
pBBR1p264-oprF-FL2-phoA	pBBR1p264-oprF-ST derivative with oprF-ST removed, replaced by oprF-FL2 and FL1-phoA encoding a (GGGGS)2 flexible linker	This study	
pBBR1p264-oprF-FL3-phoA	pBBR1p264-oprF-ST derivative with oprF-ST removed, replaced by oprF-FL2 and FL2-phoA encoding a (GGGGS)3 flexible linker	This study	
pBBR1p264-oprF-RL1-phoA	pBBR1p264-oprF-ST derivative with oprF-ST removed, replaced by oprF-RL1 and RL1-phoA encoding a (EAAAK)1 rigid linker	This study	
pBBR1p264-oprF-RL2-phoA	pBBR1p264-oprF-ST derivative with oprF-ST removed, replaced by oprF-RL2 and RL1-phoA encoding a (EAAAK)2 rigid linker	This study	
pBBR1p264-oprF-RL3-phoA	pBBR1p264-oprF-ST derivative with oprF-ST removed, replaced by oprF-RL2 and RL2-phoA encoding a (EAAAK)3 rigid linker	This study	
Primers	
oprF_F	ATGGAATTCAGGAGGTAATATTTatgaaactgaagaacaccttaggc	EcoRI	
oprF_R	ATCGTACGTAACTACCgacgttgtcgcaaacgccgtc	Eco105I	
phoA_F	AATTTACGTAcctgttctggaaaaccggg	Eco105I	
phoA_R	ATATAAGCTTtcatttcagccccagagcggc	HindIII	
CL-phoA_F	ATATTACGTAATCGACGGCCGCGGCTCCcctgttctggaaaaccgg	Eco105I	
oprF-FL1_R	AGAGGATCCGCCGCCGCCgacgttgtcgcaaacgcc	BamHI	
oprF-FL2_R	TATGGATCCGCCGCCGCCCGAGCCGCCGCCGCCgacgttgtcgcaaacgcc	BamHI	
oprF-RL1_R	CTTGGCGGCCGCTTCgacgttgtcgcaaacgcc	NotI	
oprF-RL2_R	CTTGGCGGCCGCTTCCTTCGCCGCGGCTTCgacgttgtcgcaaacgcc	NotI	
FL1-phoA_F	GGCGGCGGCGGATCCcctgttctggaaaaccgg	BamHI	
FL2-phoA_F	GGCGGCGGCGGATCCGGCGGCGGCGGCTCGcctgttctggaaaaccgg	BamHI	
RL1-phoA_F	GAAGCGGCCGCCAAGcctgttctggaaaaccgg	NotI	
RL2-phoA_F	GAAGCGGCCGCCAAGGAAGCCGCGGCGAAGcctgttctggaaaaccgg	NotI	
Note:

a The annealing portion of primers are shown in lowercase, synthetic additions uppercase, restriction sites 2 underlined, linker sequences italicized, and ribosomal binding site bolded.

Construction of a surface display system

The sequence encoding the native signal peptide plus the first 188 amino acids of OprF (OprF188) was amplified from P. aeruginosa genomic DNA using primers oprF_F and oprF_R, containing extended EcoRI and Eco105I restriction sites, respectively. The resulting oprF188 amplicon was digested with these enzymes and ligated into similarly-cut pBBR1p264-ST and pBBR1p452-ST to produce pBBR1p264-oprF-ST and pBBR1p452-oprF-ST (Table 1). The gene encoding PhoA was amplified from E. coli genomic DNA using primers phoA_F and phoA_R, containing extended Eco105I and HindIII sites, respectively. Primer phoA_F was designed to exclude the native periplasmic signal sequence of PhoA. The phoA amplicon was digested with Eco105I and HindIII and ligated in-frame into similarly-cut vectors to produce pBBR1p264-oprF-phoA and pBBR1p452-oprF-phoA. Plasmids were transformed into E. coli and transformants were screened by colony PCR and confirmed by sequencing. Plasmids were then transformed into G. oxydans by electroporation (Kallnik et al., 2010; Kosciow et al., 2016; Kostner et al., 2013). Briefly, 100 mL of electroporation medium was inoculated with an overnight culture of G. oxydans and grown to an OD600 nm of 0.8–1.0. The culture was placed on ice for 20 min, and chilled cells were harvested by centrifugation at 2,000g for 10 min at 4 °C. Cells were washed three times with 50–100% volume of 1 mM HEPES, with centrifugation at 4,000g for 10 min at 4 °C. Pellets were resuspended with HEPES and combined into a final volume of 800 μL, to which 200 μL of glycerol was added. Cells were either used immediately or aliquoted, flash-frozen, and stored at −80 °C. For electroporation, 40 μL of electrocompetent cells were combined with 1 μL of plasmid DNA and pulsed with a field strength of 22 kV/cm using a BioRad MicroPulser. After a 6–16 h outgrowth in electroporation medium, transformed cells were plated on YGC agar containing kanamycin and cefoxitin. To incorporate a cleavable linker (CL) into the OprF188-PhoA fusion protein, a 5′-extended version of the PhoA gene, CL-phoA, was amplified from E. coli genomic DNA using primers CL-phoA_F and phoA_R. Primer CL-phoA_F encoded the amino acid sequence, Ile-Asp-Gly-Arg, recognized by Factor Xa protease (Nagai & Thogersen, 1987; Terpe, 2003). The CL-phoA amplicon was cut with Eco105I and HindIII and ligated into similarly-cut pBBR1p452-oprF-ST to produce pBBR1p452-oprF-CL-phoA (Table 1).

Construction of a linker library for surface display

A library of flexible (FL) and rigid (RL) linkers was assembled similar to Li et al. (2016). The inserts, oprF-FL1, oprF-FL2, oprF-RL1, and oprF-RL2, were amplified from the plasmid, pBBR1p264-oprF-ST, using the forward primer, oprF_F, and the respectively-named reverse primers (oprF-FL1_R, oprF-FL2_R, oprF-RL1_R, and oprF-RL2_R) containing either BamHI or NotI sites. The inserts, FL1-phoA, FL2-phoA, RL1-phoA, and RL2-phoA were amplified from the E. coli genome using the reverse primer, phoA_R, and the respectively-named forward primers (FL1-phoA_F, FL2-phoA_F, RL1-phoA_F, and RL2-phoA_F) containing either BamHI or NotI sites. The linker system inserts were cut with their corresponding restriction enzymes—all oprF188 inserts were cut with EcoRI, all phoA inserts were cut with HindIII, all FL inserts were cut with BamHI, and all rigid linker (RL) inserts were cut with NotI. In parallel, vector pBBR1p264-oprF-ST was cut with EcoRI and HindIII. The linker library inserts were ligated into a linearized vector in a combinatorial fashion to produce six plasmids comprising the linker library: pBBR1p264-oprF-FL1-phoA, pBBR1p264-oprF-FL2-phoA, pBBR1p264-oprF-FL3-phoA, pBBR1p264-oprF-RL1-phoA, pBBR1p264-oprF-RL2-phoA, and pBBR1p264-oprF-RL3-phoA (Table 1).

Phosphatase assays

Alkaline phosphatase assays were done using a modified method of Kosciow et al. (2014). Experimental cultures were inoculated 1:100 from overnight cultures and grown to mid-late exponential phase (OD600 of 0.9–1.1 for E. coli and 0.6–0.9 for G. oxydans). Forty microliters of cells were mixed with 160 μL of substrate buffer (1M Tris base, 10 mM MnSO4, 10 mM ZnSO4, 1.25 mM p-nitrophenylphosphate, pH 8.0) in a 96-well microplate. The reactions were incubated with shaking for 60 min at 30 °C and the change of absorbance was monitored at 405 nm in a BioTek EL808 plate reader. Phosphatase activity was reported as absorbance change per hour, normalized by the optical density of the bacterial culture, ΔA405/(h × OD600).

PhoA localization assay

Localization of the passenger protein was conducted using a modified method Jiang & Boder (2010). YM broth was inoculated 1:100 with an overnight culture of G. oxydans harboring plasmid pBBR1p452-oprF-CL-phoA and grown to mid-late exponential phase and 500 μL was centrifuged at 2,000g for 5 min. The pellet was resuspended in 200 μL of Factor Xa buffer (100 mM NaCl, 20 mM Tris-base, 2 mM CaCl2, pH 8.0) and Factor Xa protease was added to a final concentration of 20 ng/μL. Samples were incubated overnight at 23 °C with shaking. After incubation, samples were pelleted at 2,000g for 5 min, and supernatant was transferred to a new microcentrifuge tube. The supernatant was centrifuged at 16,100g for 2 min to remove intact cells and 40-μL aliquots were used to quantify phosphatase activity as described above.

Growth behavior

Yeast mannitol broth was inoculated to an OD600 of 0.05 from overnight cultures of G. oxydans strains and 1 mL of inoculated broth was added to a 24-well microplate. The plate was incubated at 30 °C with shaking at 150 rpm for up to 24 h, and absorbance was monitored at 595 nm every 5 min in a Flurostar Optima plate reader (BMG Labtech, GmbH, Ortenberg, Germany).

Statistical analysis

R Studio was used to perform statistical analyses and to generate box-and-whisker plots, strip charts, and growth curve graphics (R Core Team, 2017). Data were analyzed by performing an analysis of variance and a post-hoc Tukey’s HSD test (q = 0.05) unless otherwise noted. The R packages used in this study were dplyr, ggplot2, growthcurver, plyr, multcomp, and reshape2 (Hothorn, Bretz & Westfall, 2008; Sprouffske & Wagner, 2016; Wickham, 2007, 2009, 2011; Wickham et al. 2017).

Results

Surface display in G. oxydans

To enable surface display in acetic acid bacteria, a truncated version of OprF (OprF188) was tested for its ability to localize the PhoA reporter enzyme to the cell surface of G. oxydans. PhoA was translationally fused to the C-terminal end of OprF188, and the resulting OprF188-PhoA fusion protein was produced via two expression vectors, one containing a high-strength promotor (p264) and the other containing a moderate-strength promotor (p452) (Kallnik et al., 2010). As a preliminary test, these surface display constructs were expressed in E. coli and phosphatase activity was quantified in a whole-cell assay (Fig. 1). The OprF188-PhoA surface display systems produced statistically significant absorbance changes compared to strains expressing the anchor protein alone when using both the high- (q < 0.001) and moderate-strength (q < 0.001) promoters, p264 and p452, respectively. Enzymatic rates were approximately sixfold higher in the p264-oprF-phoA system compared to the p452-oprF-phoA system.

Figure 1 OprF as a surface display anchor in E. coli.

The OprF188-PhoA fusion protein was produced in E. coli using the high-strength (p264) and moderate-strength (p452) promoters. Phosphatase activity was measured in whole-cell reactions. Respective strains producing only the anchor peptide fused to a Strep-tag (ST) served as negative controls. Rate of PhoA activity was monitored as ΔA405 nm/(h × OD600 nm). Letters above the plot denote statistical groups determined by an ANOVA and post-hoc Tukey’s HSD test.

In G. oxydans, the p264-oprF-phoA strain produced a mean absorbance change of 0.39/(h × OD600), corresponding to a volume activity of 3.21 mU/(mL × OD600), which was significantly greater than basal activity (q < 0.001) (Fig. 2). This level of activity is approximately threefold lower than that observed when the same construct was expressed in E. coli. The p452-oprF-phoA strain produced a mean absorbance change of 0.20/(h × OD600) and a volume activity of 1.65 mU/(mL × OD600), which was significantly higher than that of the control (q < 0.001). In contrast to the high-strength promotor, there was no reduction in activity when OprF188-PhoA expression was directed by the moderate-strength promoter in G. oxydans rather than E. coli.

Figure 2 OprF as a surface display anchor in G. oxydans.

The OprF188-PhoA fusion protein was expressed in G. oxydans using the high-strength (p264) and moderate-strength (p452) promoters. Phosphatase activity was measured in whole-cell reactions. Respective strains producing only the anchor peptide fused to a Strep-tag (ST) served as negative controls. Rate of PhoA activity was monitored as ΔA405 nm/(h × OD600 nm). Letters above the plot denote statistical groups determined by an ANOVA and post-hoc Tukey’s HSD test.

To verify that PhoA was exposed to the extracellular space, a CL was incorporated into the fusion protein sequence. The resulting construct, p452-oprF-CL-phoA, was transformed into G. oxydans and an assay to localize PhoA was done (see Methods). The mean level of phosphatase activity observed in the supernatant from cells treated with Factor Xa protease was higher than that from untreated cells (two-sample t-test, p < 0.001) (Fig. 3). These data suggest that PhoA was released into the supernatant after cleavage, indicating that PhoA was attached to the outer membrane and oriented toward the medium.

Figure 3 Cleavable linker assay.

To confirm proper localization of the passenger enzyme, a cleavable linker motif was incorporated into the OprF188-PhoA surface display system. Whole G. oxydans cells were treated with Factor Xa protease and the resulting supernatant (SN) was assayed for phosphatase activity. Rate of PhoA activity was monitored as ΔA405 nm/(h × OD600 nm).

Linker system optimization

To optimize surface display in acetic acid bacteria, a library of linkers was integrated into the OprF188-PhoA surface display systems. First, overhang PCR was used to add linker building blocks to oprF188 and phoA. These building blocks encoded for either flexible or RLs and contained either BamHI or NotI restriction sites, respectively. The modified oprF188 inserts contained abridged linker sequences at their 3′ ends and the modified phoA inserts contained abridged linker sequences at their 5′ ends. Therefore, complete linker sequences resulted from ligation of compatible insert dyads. The building blocks were assembled into the expression vector that showed the highest activity to create six constructs encoding fusion proteins varying in linker composition and linker length: Three containing FLs (FL1, FL2, and FL3) composed of a (GGGGS)1–3 motif, and three containing RLs (RL1, RL2, and RL3) composed of the (EAAAK)1–3 motif.

The linker library was first expressed in E. coli and PhoA activity was quantified (Fig. 4). The FLs had slight negative effects on enzyme activity. While there was no difference between the control lacking a linker and the fusion protein containing the small FL (FL1) (q = 0.939), the medium (FL2), and large (FL3) FLs led to a slight but statistically significant decrease in activity (q < 0.001 for each). In contrast, the small (RL1) and medium (RL2) RLs did not have any effect on activity (q = 0.071 and 0.969, respectively). The large RL (RL3) led to a slight but statistically significant increase in enzymatic activity when compared to the control (q < 0.001). However, there was no difference between the RL1 strain and the RL3 strain (q = 0.692), suggesting that the higher activity observed for the RL3 strain was likely not biologically significant.

Figure 4 The effects of linkers on biocatalysis in E. coli.

To determine the effects of linkers on the surface display system, a library of flexible (GGGGS)1–3 and rigid (EAAAK)1–3 linkers was integrated into the OprF188-PhoA fusion protein and PhoA activity was measured. Rate of PhoA activity was monitored as ΔA405 nm/(h × OD600 nm). Letters above the plot denote statistical groups determined by an ANOVA and post-hoc Tukey’s HSD test.

The linker systems had more pronounced effects on PhoA activity in G. oxydans compared to the subtle effects of the flexible and RLs in E. coli (Fig. 5). While the small FL (FL1) and the medium RL (RL2) had no effect on activity when compared to the control lacking a linker (q = 0.967 and q = 0.765, respectively), the large RL (RL3) led to a 32% reduction in activity (q < 0.001). Interestingly, the small RL (RL1) drastically improved PhoA activity (q < 0.001), having a 69% increase in biocatalysis that corresponded to a volume activity of 5.42 mU/(mL × OD600). Attempts to transform plasmids p264-oprF-FL2-phoA and p264-oprF-FL3-phoA into G. oxydans were unsuccessful, suggesting that the medium (FL2) and large (FL3) FLs were toxic at high expression.

Figure 5 The effects of linkers on biocatalysis in G. oxydans.

A linker library consisting of flexible (GGGGS)1 and rigid (EAAAK)1–3 linkers was integrated into the OprF188-PhoA fusion protein and expressed in G. oxydans and the phosphatase activity was measured. Rate of PhoA activity was monitored as ΔA405 nm/(h × OD600 nm). Letters above the plot denote statistical groups determined by an ANOVA and post-hoc Tukey’s HSD test.

Effect of surface display on growth of G. oxydans

To characterize the effects of protein production and surface engineering on the growth behavior of G. oxydans cells, a method was developed to follow the growth of recombinant G. oxydans strains using standard 24-well microplates, allowing automated monitoring of cell growth in a plate reader. Strains containing OprF188 surface display systems were compared to wildtype G. oxydans 621H growth (Fig. 6). When the high-strength p264 promoter was used, production of the OprF188 anchor protein without fusion to PhoA led to a statistically significant increase in mean doubling time (97 min) compared to wildtype cells (56 min) (q < 0.001) (Fig. 7). However, when the OprF188-PhoA fusion was expressed, the doubling time (52 min) was not different than that of wildtype cells (q = 0.603). The presence of the small FL (FL1) increased the doubling time (82 min) compared to the strain expressing the OprF188-PhoA fusion without a linker (q < 0.001). Conversely, the presence of the small RL (RL1) decreased doubling time (27 min) compared to the strain lacking the linker (q < 0.001) and to wildtype cells (q < 0.001). However, the lag time for this strain was approximately 8 h longer than that of wildtype cells (Fig. 6). The medium RL (RL2) did not affect doubling time (58 min) compared to the no-linker control strain (q = 0.166) or wildtype cells (q = 0.952) (Fig. 7). Lastly, the large RL (RL3) increased doubling time to 121 min. Generally, the lag time was longer for recombinant strains and the final optical density was lower than that of wildtype G. oxydans, except for the p264-oprF-RL1-phoA containing strain (Fig. 6) that also produced the highest PhoA activity. Interestingly, doubling time was inversely proportional to PhoA activity (Figs. 5 and 7, respectively), suggesting that recombinant PhoA positively contributed to the growth of G. oxydans.

Figure 6 Growth behavior of recombinant G. oxydans strains.

Cells were cultured in 24-well microplates and the growth of recombinant G. oxydans 621H strains was compared to that of wildtype. Solid line, mean optical density; ribbon, 95% CI; n = 3.

Figure 7 Doubling time of G. oxydans strains.

Doubling time was calculated using the growthcurver R package. Letters above the plot denote statistical groups determined by an ANOVA and post-hoc Tukey’s HSD test.

Discussion

In this study, the gene encoding PhoA was fused to OprF188 to test the ability of the anchor peptide to transport recombinant enzymes to the cell surface of acetic acid bacteria. The resulting constructs were expressed in the model acetic acid bacterium G. oxydans and biocatalysis was quantified in whole-cell reactions. Based on enzymatic activity, OprF188 localized active PhoA to the cell surface, regardless of expression level. Nascent PhoA has no activity in the cytoplasm as it does not fold properly unless secreted extracellularly where disulfide bonding takes place (De Geyter et al., 2016; Ehrmann, Boyd & Beckwith, 1990; Hoffman & Wright, 1985; Manoil & Beckwith, 1985; Michaelis et al., 1983). The native periplasmic signal sequence was removed from PhoA before it was fused to OprF188, which contains an outer membrane signal peptide and is known to localize to the outer membrane (Lee et al., 2005; Sugawara, Nagano & Nikaido, 2012). The OprF188-PhoA fusion was produced using two expression vectors designed for protein production in G. oxydans, one containing a high-strength p264 promoter, and the other a moderate-strength p452 promoter. There was a twofold difference in phosphatase activity produced by G. oxydans between the two promoters. This mirrors the threefold difference in enzymatic activity reported previously (Kallnik et al., 2010).

Because PhoA is an innately periplasmic enzyme, its location needed to be verified to show that OprF188 can correctly target recombinant enzymes to the outer leaflet of the outer membrane in G. oxydans. To this end, a CL was integrated between OprF188 and PhoA as was done previously to validate another surface display systems (Jiang & Boder, 2010). If PhoA was exposed at the cell surface via OprF188, then PhoA should be removed from the outer membrane following treatment with Factor Xa protease, increasing phosphatase activity in the medium. The CL assay demonstrated that the level of phosphatase activity in the treated supernatant fractions was indeed higher than that in the untreated supernatant fractions (p < 0.001), which suggests that PhoA was present at the outer leaflet of the outer membrane.

In this study, we used a C-terminal truncated version of OprF, OprF188, containing 188 amino acids from the N-terminus of the original protein in its mature form. Valine 188 was previously determined to be the optimal fusion site for passenger proteins (Lee et al., 2005). This anchor protein was previously used to localize a 49.9 kDa lipase, which is comparable to the size of the 47 kDa PhoA monomer (Bradshaw et al., 1981), to the outer membrane of E. coli (Lee et al., 2005) and Pseudomonas putida (Lee, Lee & Park, 2005). The first 24 amino acids of nascent OprF188 encodes a Sec-dependent signal peptide. OprF is an unusual outer membrane protein in that it is comprised of two domains: An N-terminal domain that forms a small, eight-stranded β-barrel, and a C-terminal globular domain that associates with the cell wall in the periplasm (Bodilis & Barray, 2006; Sugawara, Nagano & Nikaido, 2012). Mature OprF188 consists of only the β-barrel domain with a C-terminal extracellular loop (Lee et al., 2005). OprF is also unusual because the β-signal that is recognized by the β-barrel assembly complex to facilitate outer membrane integration, is located internally at the end of the β-barrel domain, rather than C-terminally (Gessmann et al., 2014; Sugawara, Nagano & Nikaido, 2012). Thus, truncated OprF188 maintains its β-signal, allowing proper membrane integration (Lee et al., 2005; Sugawara, Nagano & Nikaido, 2012).

Interestingly, there have been many unsuccessful attempts to target PhoA to the cell surface of E. coli using E. coli outer membrane proteins as anchors. PhoA was fused to the ferrichrome outer membrane transporter FhuA (Coulton, Reid & Campana, 1988). While PhoA was associated with the outer membrane, it was not necessarily exposed to the outer surface. Similarly, the ferric enterobactin outer membrane transporter FepA localized PhoA to the outer membrane of E. coli, but it was periplasmically oriented (Murphy & Klebba, 1989). When a PhoA was fused to a lipoprotein-OmpA hybrid, again the fusion protein was associated with the outer membrane but PhoA existed exclusively in the periplasm (Stathopoulos, Georgiou & Earhart, 1996). Close inspection of these studies reveals that the β-signal was removed from FhuA, FepA, and OmpA upon C-terminal truncation. Therefore, deletion of the β-signal was likely sufficient to prevent proper outer membrane insertion by the β-barrel assembly complex in those studies. The use of OprF188 as an anchor to successfully display proteins on the cell surface in the current study and by others suggests that it is broadly functional as a surface display anchor protein in gram-negative bacteria, likely due to the preservation of the β-signal (Lee et al., 2005; Lee, Lee & Park, 2005).

Despite their influence on fusion protein activity, the effect of linkers on bacterial surface display has not previously been investigated. While linkers are sometimes included in the design of surface display systems, their inclusion is rarely made explicit and even fewer studies offer any explanation as to why a particular linker sequence was chosen. Yet linker sequences can play a vital role in the design of fusion proteins, affecting protein folding and production efficiency, and influencing enzyme activity (Chen, Zaro & Shen, 2013; Li et al., 2016). Here, a library of linkers varying in both composition and length were integrated into a surface display system to explicitly test the effects of linkers on biocatalysis at the cell surface of two bacterial species. Two types of linkers were investigated in this study, FLs and RLs. FLs are composed of small, polar amino acids such as glycine and serine and form random coils, such as the (GGGGS)1–3 linker used here. RLs are often α-helical, such as the (EAAAK)1–3 linker used here (Chen, Zaro & Shen, 2013; Li et al., 2016). FLs provide passive separation and permit interaction between the components of fusion proteins, while RLs maintain a set distance and can prevent interaction between fusion protein domains (Li et al., 2016). The linker library was first expressed in E. coli and, generally, the presence of linkers did not dramatically influence product yield. In contrast, the linkers caused pronounced effects on biocatalysis in G. oxydans. Fusion proteins containing the medium (FL2) and large (FL3) FLs—10 and 15 amino acids in length—were likely toxic to G. oxydans, as positive transformants were not obtained after multiple attempts. The fusion proteins containing the small flexible (FL1) and medium rigid (RL2) linkers had no influence on biocatalysis, whereas the large rigid (RL3) linker was deleterious. Optimization of the surface display system was achieved with the small RL (RL1), consisting of a single EAAAK pentapeptide, which improved phosphatase activity by 69%.

The disparate results of the linkers on E. coli and G. oxydans suggest that the effects of linkers on biocatalysis at the cell surface may be species-specific. E. coli and G. oxydans are phylogenetically distant, belonging to the Gammaproteobacteria and Alphaproteobacteria, respectively. There are likely differences in the composition of their outer membranes and thus the environment at the cell surface of these bacteria. It should be noted that the outer membrane composition of G. oxydans and other acetic acid bacteria remains largely uncharacterized, limiting direct comparison. Nevertheless, these differences may also be why phosphatase activity was overall higher in E. coli than in G. oxydans. Differences in promoter strength and/or efficiency of secretion (Choi & Lee, 2004) or outer membrane protein assembly by the β-barrel assembly complex (Browning et al., 2015; Sikora et al., 2017) may also play a role in the differences in the activities observed between E. coli and G. oxydans. Additionally, the differences in linker effects could also be passenger protein-specific. Overall, these results suggest that linker optimization is an important consideration for each surface display system used and for the specific host organism.

To determine the effects of protein production and surface engineering on the growth of G. oxydans, a novel method was developed for semi-high-throughput culturing of G. oxydans. The strain containing plasmid p264-oprF-ST had significantly slower growth compared to wildtype G. oxydans 621H. It is possible that this growth defect was caused by the metabolic burden of protein production. Alternatively, the production of the OprF188 peptide may have overwhelmed secretory machinery, preventing sufficient amounts of required proteins from being processed. It is also possible that integration of the OprF188 anchor destabilizes outer membrane integrity. Interestingly, the p264-oprF-phoA strain did not exhibit a growth defect, suggesting that PhoA may enable recovery of the OprF188 phenotype. It may be that high expression of PhoA allowed cells to scavenge more phosphate from the medium, possibly allowing recovery from outer membrane lipid defects. Indeed, the small RL (RL1) not only increased PhoA activity, but also improved the growth rate with concomitant increased lag time. PhoA activity seemed to be inversely proportional with doubling time, suggesting that inorganic phosphate availability may contribute to the observed growth phenotypes.

Conclusion

Surface display is potentially a powerful tool to enable metabolic engineering of G. oxydans and other industrially important acetic acid bacteria. G. oxydans currently has a limited ability to grow on disaccharides and polysaccharides, and instead relies on relatively expensive monomeric feedstocks (Kosciow et al., 2016). The ability to express enzymes at the outer membrane could be used to improve current bioprocesses by broadening the substrate range of this bacterium. Because this technique utilizes export machinery found in all gram-negative bacteria, the surface display constructs could be expressed in otherwise wildtype cells. Previously, a recombinant exoenzyme with a periplasmic signal peptide was efficiently released from cells only when a knockout strain of G. oxydans with a leaky outer membrane phenotype was used (Kosciow et al., 2016). Surface display could also be used in conjunction with immobilized G. oxydans cells to create stable bioreactors, such as those used in the production of dihydroxyacetone (Dikshit & Moholkar, 2016). In conclusion, a novel molecular tool for strain improvement of acetic acid bacteria was produced, and the OprF188 surface display system described herein is a significant first step toward outer membrane engineering of G. oxydans. Such molecular tools will enable engineering of this unique bacterium to improve and expand its ability to produce value-added products.

Supplemental Information

Supplemental Information 1 Raw data used to generate statistics.

Raw data for Figs. 1–7.

Click here for additional data file.

Supplemental Information 2 R scripts used to generate figures and statistics.

Click here for additional data file.

We thank Dr. Kyoungtae Kim and Dr. Laszlo Kovacs for lending their expertise and providing feedback on this project.

Additional Information and Declarations

Competing Interests

Author Contributions

Data Availability

The authors declare that they have no competing interests.

Marshal Blank conceived and designed the experiments, performed the experiments, analyzed the data, contributed reagents/materials/analysis tools, prepared figures and/or tables, authored or reviewed drafts of the paper, approved the final draft.

Paul Schweiger conceived and designed the experiments, analyzed the data, contributed reagents/materials/analysis tools, prepared figures and/or tables, authored or reviewed drafts of the paper, approved the final draft.

The following information was supplied regarding data availability:

The raw data that was used to generate statistics/error bars and corresponding R code for each figure are available as Supplemental Files.

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
