# Peer review of "Surface display for metabolic engineering of industrially important acetic acid bacteria"

_PeerJ, doi:10.7717/peerj.4626_

## Round 0.1 · original submission · Minor Revisions

Both reviewer agree that your work is of high quality and broad interest. Their (minor) comments should help to improve the presentation of your manuscript.

·

Basic reporting

In this manuscript, the authors provide convincing evidence with statistical support that their surface display system is sufficient in producing recombinant enzymes at the outer membrane of Gluconobacter oxydans. This is of particular interest in the field, because of the difficulties associated with engineering G. oxydans membranes to produce useful proteins that can directly interact with the surrounding environment. It is my opinion, this manuscript is acceptable for publication with a few minor changes, which are listed below.

Minor concerns:
Both lines 35-37 of abstract and lines 80-83 state (I paraphrase here) that the expression of extracytoplasmic enzymes is difficult in the inner membrane because the membrane is crowded with proteins. This statement underlines the importance of their work and why the development of a surface display system in Gluconobacter oxydans is a potential bioengineering solution. It would help their case to cite the “crowdedness of the membrane.” For example, see Guigas and Weiss 2016 Biochim Biophys Acta (PMID: 26724385).

Line 198: Samples were incubated at 23 C for how long?

Line 248: I would remove the word “significantly.” Although the t-test shows a clear difference, there is an apparent overlap between samples in Fig. 3, and this is reflected in their ANOVA analysis.

Line 336: The same comment for line 248, remove “significantly.”

Lines 416-418: Although it is not required for this manuscript, I am curious if the authors have tried adding more phosphate to the media and measured the growth. Does additional phosphate rescue the slowed growth they observed when producing PhoA?

Figures 1-5 legends: It would be helpful if the units (deltaA405/hxOD600) for the phosphatase activity were included in the legend.

Experimental design

No comment

Validity of the findings

No comment

Additional comments

No comment

Reviewer 2 ·

Basic reporting

There are minor comments to the writing, detailed in the general comments to the author section.

Experimental design

No comment

Validity of the findings

Some discussion might be required, see general comments section.

Additional comments

This is an interesting piece of work where the authors make a proof of concept for surface display in an effort to potentiate the abilities of Gluconobacter oxydans as producer of industrial metabolites. The authors fuse a N-terminal of OprF from Pseudomonas aeruginosa to a alkaline phosphatase (PhoA) from E. coli under the control of strong and moderate promoters. The authors also assay the contribution of flexible and rigid linkers. They make supernatant phosphatase assays in E. coli and G. oxydans. Concluding that for G. oxydans the genetic construction that works better includes the use of the strong promoter and a short rigid linker fused to OprF and PhoA.
I think the work could be of interest for the readers of the journal and recommend their publications, I just suggest the authors could attend the following points:
1. The abstract could be improved stating that for G. oxydans the strong promoter is better, and resuming how is the whole genetic construction that works better.
2. In the discussion section the authors could argue why the phosphatase activity in E. coli is superior than the best achieved in G. oxydans.
3. On line 114, there is repeated the word “in”. On lines 324-326 please revise the words promotor-promotors.

---

## Round 0.2 · accepted · Accept

You have adequately addressed all reviewers' comments. Good luck with the promotion of your interesting work!

#